# Explanatory Change Detection in Financial Markets by Graph-Based Entropy and Inter-Domain Linkage

**DOI:** 10.3390/e24121726

**Published:** 2022-11-25

**Authors:** Yosuke Nishikawa, Takaaki Yoshino, Toshiaki Sugie, Yoshiyuki Nakata, Kakeru Itou, Yukio Ohsawa

**Affiliations:** 1Department of Systems Innovation, School of Engineering, The University of Tokyo, 7-3-1, Hongo, Bunkyo-ku, Tokyo 113-8656, Japan; 2Nissay Asset Management Corporation, Marunouchi Building, 1-6-6, Marunouchi, Chiyoda-ku, Tokyo 100-8219, Japan

**Keywords:** change detection, graph entropy, financial market

## Abstract

In this study, we analyzed structural changes in financial markets under COVID-19 to support investors’ investment decisions. Because an explanation of these changes is necessary to respond appropriately to said changes and prepare for similar major changes in the future, we visualized the financial market as a graph. The hypothesis was based on expertise in the financial market, and the graph was analyzed from a detailed perspective by dividing the graph into domains. We also designed an original change-detection indicator based on the structure of the graph. The results showed that the original indicator was more effective than the comparison method in terms of both the speed of response and accuracy. Explanatory change detection of this method using graphs and domains allowed investors to consider specific strategies.

## 1. Introduction

In financial markets with complex price movements, it is important for investors to understand the changes in the market and make appropriate responses and investment decisions. In particular, investors should avoid the pain caused by major market changes during major events, such as COVID-19; many studies have been conducted on the market during the spread of COVID-19 [1,2,3].

Although there have been studies on detecting changes in financial markets [4,5,6], it is important to explain why a change has occurred: not just to detect it, but to respond appropriately to the change and prepare for similar major changes in the future. Specifically, consider the case in which a drop in the price of stocks in Industry A leads to a drop in the price of stocks in Industry B. The explanation would identify the stocks that should be avoided at present as stocks related to Industry B, and when the next drop in the price of stocks in Industry A occurs, attention can be paid to Industry B as well. Another example is the case in which investors know whether the decline in Stock A is due to long-term or short-term reasons. In this case, they know if this decline is something that will soon be recovered from, and this can prevent them from acting inappropriately due to anxiety.

Graph visualization is an excellent method for generating strategies based on the analysis of data because structural changes are visible. Ohsawa (2018) discovered new business opportunities by explaining changes through graphical visualization of the structure of the co-occurrence of events.

Research on change detection for graphs has been conducted in various ways [7,8,9]. In financial analysis, there have been studies in which the structural changes in stock markets were represented graphically [10,11,12,13,14]. Huang (2009) and Xiu (2021) conducted an analysis based on the characteristics of daily stock-related graphs, wherein they analyzed the graphs as a whole rather than focusing on individual assets or their connections, which made it difficult for investors to generate concrete ideas.

Based on the above background, this study uses the approach of detecting market changes, explaining them by visualizing market-correlation graphs, and dividing them into finance domains for detailed analysis to support investors’ investment decisions. To detect changes, we focused on the impact of the most recent significant event, COVID-19. In addition to the indicators looking at the graph as a whole, to focus on changes in market conditions and to provide an explanation for such changes, we looked at the graph divided. We conducted a detailed analysis in the hope that the connections in specific areas could explain the cause of the change.

We will compare this study with the study of financial markets under COVID-19 [15,16]. Hatipoğlu (2021) analyzed the economic network under COVID-19 using the minimum spanning tree. However, it used more than one year of data to form the network, and the minimum spanning tree is difficult to compute. The daily graph in our study consists of data with a window width of 10 days, which was faster to calculate and more suited to investor needs. Graphs are generated daily, allowing information to be collected highly frequently. Akhtaruzzaman analyzed the stock returns of countries during COVID-19 by looking at the correlation coefficient between China and other countries. However, correlation coefficients can only measure a one-to-one relationship. Our method used clusters of graphs to capture shifts in the overall structure of the market that are difficult to grasp in a one-to-one analysis.

Section 2 describes graph-based entropy, an entropy measure calculated from the cluster structure of graphs, and its development indicator, inter-domain linkage. Section 3 describes the experimental methods and hypotheses of this study. Section 4 describes the results of the experiment, Section 5 discusses the results, and Section 6 concludes the paper and discusses future prospects.

## 2. Related Research

### 2.1. Graph-Based Entropy

Graph-Based Entropy (GBE) (1) [17] is an index that detects when the structure of a graph has changed due to external factors using entropy calculated from clusters. The ups and downs of the entropy of a graph can numerically capture structural changes in the graph reflecting the latent dynamics underlying the original data, quantified along with the visualization of the graph. GBE is an extension of entropy and a measure of clutter in graphs. In a graph, there is more clutter and more information when there are multiple clusters than when all nodes belong to a single cluster. Similarly, in financial markets, when all stocks are moving in the same direction, there is less clutter and less information than when they are all moving in discrete price movements. By calculating entropy from the ratio of nodes in a cluster to the total, GBE captures the variation in clutter and information content of the overall financial graph, thus capturing changes in the structure of the market. Ohsawa (2018) applied this measure to a graph representing items purchased together, created from supermarket POS data, to discover shifts in the cluster structure of product groups and use it for product sales strategies. We have applied this measure to the relevant graphs of the financial market, of which the transition provides a broad picture of the structural changes in the market [18]. For example, the effects of COVID-19 affected the causalities in the financial market (e.g., the local influence of the US market on the Japanese market is no longer special because all countries may go into a downtrend). In this study, GBE was applied to the graph of the relationship between assets in the financial market in order to detect changes in the structure of the market and to explain the structural changes in the market through visualization to help investors make investment decisions. A graph of financial assets as nodes, connected by edges to those with close price movements, is created daily, and edges connected by edges are treated as clusters.
(1)$$GBE=-\underset{j}{\sum }p\left(cluster_{j}\right)logp\left(cluster_{j}\right)\;p\left(cluster_{j}\right)=\frac{V\left(cluster_{j}\right)}{V}$$

*p(clusterj)*: percentage of all nodes belonging to cluster *j* from all nodes;

*V*: the number of all nodes.

### 2.2. Inter-Domain Linkage

Inter-domain linkage is an extension of GBE and is a measure of the variability of the data in the same cluster. It increases when there are many clusters with nodes that straddle different domains when domains are set up in the graph. A domain is a grouping of the nodes of a graph; for example, nodes grouped together in the same industry are defined as one domain. When data with similar value movements are grouped into domains, the occurrence of clusters that straddle these domains is considered to be the result of some background events that affected different groups that were not sufficiently covariate to be clustered together. By dividing the data into domains and examining the degree of connectivity, it is possible to consider background events that would be difficult to notice by looking at the entire graph. In this study, we set the stock price index and the government bond interest rate as two domains, as shown in Figure 1. By examining the inter-domain linkage, we aimed to provide a more detailed explanation of the changes, such as how the structural changes in the graph were made across the two domains.
(2)$$I\left(G\right)=\sum \left[C_{k}∊B\right]P\left(C_{k}\right)\;*\;\sum \left[S_{i}\subset G\right]\{-P(S_{i}|C_{k})logP(S_{i}|C_{k})\}$$

*G*: a graph.

*S_i_*: *i*-th domain of *G.*

*B*: clusters in *G* that straddle domains.

## 3. Experiment

### 3.1. Purpose and Hypothesis

The purpose of this study is to represent the structure of the financial market in daily correlation graphs and to apply GBE and inter-domain linkage to detect and explain changes in the market.

To confirm that the market has changed due to an impact on Japan, we added data that would represent the Japanese market as a whole, in addition to stock price indices, to form a domain. By examining the linkage between that domain and the stock price domain, we detected spillover effects in Japan. In this study, one domain was established with government bond interest rates as data that could represent the Japanese market as a whole. Figure 1 shows hypothetical images of the evolution of the stock price index and JGB interest rate.

In Figure 1a, which represents the normal state, the assets have their own price movements, and most of them are uncorrelated. In Figure 1b, the market begins to be affected by external factors, and the price movements of some assets of the same type show a correlation, forming clusters. Finally, in Figure 1c, a cross-domain cluster occurs, which means that the market as a whole has been severely affected by a common cause to the extent that even different types of assets show correlations.

In this study, we verify whether the flow of change in the financial market is in accordance with hypothetical images, as well as detect and explain structural changes.

### 3.2. Data

To focus on the major impact of the series of events related to COVID-19 in Japan, we prepared daily data for a total of 494 days from 1 January 2019, to 20 November 2020, for 17 TOPIX-17 indexes by industry and six JGB interest rates (2-year, 5-year, 7-year, 10-year, 20-year, and 30-year) (Table A1).

### 3.3. Daily Correlation Graph

Based on the prepared daily data, we created a graph showing the daily relationships between financial market assets.

The 23 time-series data corresponding to the 23 assets were set as nodes, and each was divided with a window width of the past 10 days, standardized within the window width. We chose a window width of 10 days, which is the equivalent of 2 business weeks, because this is a sufficient number of days to calculate the distance between time series and to quickly capture changes, as mentioned in chapter 1. For all node combinations, the distance between the time series of assets within a window width was calculated using the dynamic time warping (DTW) method [19], and daily correlation graphs were created by edging combinations of nodes whose distances were lower than a specified threshold *θ*.

### 3.4. Domain

To compute the inter-domain linkage, we divided the data into two domains. To put correlated data in the same domain, we divided the data into two domains by the k-means method using daily data for the past five years (2014–2018), and the distance between each time series was calculated by DTW. As a result, the data were divided into two domains: 17 TSE stock price indices by industry, excluding the real estate index (stock price index domain), and six government bond interest rates plus the real estate index (government bond interest rate domain) (Table A1).

### 3.5. Change-Detection Indicators

To detect the point at which the structure of the financial market shifts from a normal structure to a state where changes begin to occur, we computed the following original indicator prior domain change (PDC) (3). To capture the above transition, the PDC focused on the change in GBE and inter-domain linkage during the change from Figure 1a to Figure 1b. To capture changes in the structure of the financial market from normal to anomalous conditions as quickly as possible, the indicator is designed to react quickly at the point where the clusters become larger within each domain before the graph completely crosses the domain and becomes a cluster, putting (1- *I*) in the numerator and GBE in the denominator. In a normal financial market, the PDC is low because the inter-domain linkage is low, and the GBE is high, as edges are not well connected. When a significant change that would affect the financial market as a whole occurs in the financial market, the number of edges in domains increases; therefore, the inter-domain linkage remains low, GBE goes down, and the PDC goes up. When the change reaches the entire market and more edges of the graph are connected, the PDC’s numerator is lower because the inter-domain linkage is higher. As in the above flow, the PDC is an indicator that goes up while change occurs in the market.
(3)$$PDC_{t}=\frac{1-I_{t}}{GBE_{t}}\;\left(0\leq I_{t}\leq 1\right)$$

*t*: time step.

## 4. Results

### 4.1. Changes in Daily Correlation Graphs

Of the generated daily correlation graphs with a large drop in GBE, those of Figure 2 are around late February 2020, when the threshold *θ* for edging is 0.6.

### 4.2. Change Detection Method

Figure 3 shows the original change indicator PDC; the change finder [20,21], which detects changes in time-series data; the change point [22], which detects changes in graphs; and the TOPIX TSE stock price index. The change finder is an indicator for detecting change points in a single time-series data set and uses an extension of the autoregressive (AR) model, in which current output is influenced by previous output, to learn a probabilistic model of time-series data online. This study applied the change finder to the Tokyo Stock Exchange Stock Price Index (TOPIX index), which can be considered a summary index of the TOPIX-17 index by industry, used in daily correlation graphs. The change point is a method for change detection based on how far away the principal eigenvector is from the average of the past principal eigenvectors, using the time-series data of the feature values of each node in the graph, and creating a daily correlation matrix by dividing it by the window width. In this study, the features were set to the order of the node, and the window width was set to 10 days to compute the change points. Figure 3a shows the transition of the change detection index and the overall TOPIX when the distance threshold *θ* was set to 0.6.

Figure 3b,c zooms in on January and February 2020 and May and June 2020, respectively, from the data in Figure 3a.

## 5. Discussion

### 5.1. Changes in Daily Correlation Graphs

The daily correlation graph in Section 4.1 shows that as of 12 February 2020 (Figure 2a), when the finance market was in a normal state, a small number of edges were connected, as shown in Figure 1a in the hypothetical figure. However, on 25 February 2020 (Figure 2b), when COVID-19 began to have an impact, clusters began to form among the TOPIX and JGB interest rates, as shown in Figure 1b; on 2 March 2020 (Figure 2c), the clusters became large, as shown in Figure 1c. This sequence of events is similar to that shown in the hypothetical diagram in Section 3.1 (Figure 1). It is thought to represent a flow in which the price movements of each asset, which were scattered at the beginning, were gradually affected by the COVID-19 event. Finally, all the price movements were aligned in a negative direction.

### 5.2. Change Detection

In Figure 3a of Section 4.2, the original indicator PDC, together with the comparison indicator change finder, reacted most significantly during late February 2020. This is the time when the TOPIX index fell most significantly, and compared to the change point, which is a change-detection indicator using the same graph, the PDC is able to capture the large change caused by COVID-19. Because the change point focused on only one vector at each point in time, it is believed to have been less successful in capturing the overall structural changes in complex financial markets than our method, which deals with structural changes in clusters. We believe that the PDC captured the major changes in the market due to COVID-19, as seen in Section 5.1, by the connectivity of domains between the data linked to the overall market and the stock data.

In Figure 3b, zooming in on the January/February 2020 section, the PDC did not increase on 29 January 2020, but the change finder increased. The PDC did not increase because this was a period when both interest rates and stock prices fell. However, before the important changes, such as the big drop in stock prices on 25 February 2020 and the halt in interest rates, the PDC was up, and we were able to focus on the important changes and capture them. For the change in late February 2020, the change finder was noticeably higher than the previous 10 days on 2 March 2020, but the PDC was noticeably higher than the previous 10 days on 28 February 2020, and detection was made earlier. In the late May 2020 changes in Figure 3c, the change finder was noticeably higher than the previous 10 days on 29 May 2020, but the PDC was on 28 May 2020 and reacted sharply.

The ROC curves for each of the change-detection indicators are drawn and summarized in the graphs in Figure 4. The ROC curve is a curve with the true-positive rate on the vertical axis and the false-positive rate on the horizontal axis, which is used as an evaluation index for the ability to detect a certain signal on a designed measure. Because the values of the change finder and the change point were higher at the beginning of the time series, we cut out the first 80 days for all three and used only the days after 23 April 2019 for comparison. The anomalous point to be detected is defined as the top 15% day of the TOPIX index’s percentage decline from the previous day. The ROC curve was drawn by changing the threshold value of the indicator to determine whether or not the index was anomalous. The original indicator PDC could detect large changes more accurately than the change finder and change point when the threshold for the reaction point was stricter.

We also used the benefit and false-alarm rates used by Yamanishi (2002) to evaluate the change finder. The benefit and false-alarm rates for each change-detection index are shown in Figure 5. Benefit is expressed by Equation (4) and is an index that considers how quickly the change-detection index reacts to an anomalous point. The false-alarm rate is the percentage of the detection points that reacted at a point that was not an anomalous point. PDC is considered superior to the other indicators in terms of speed of detection because the average value of the benefit is also better than the other indicators when the threshold of the reaction point is stricter.
(4)$$Benefit\left(t\right)=1-\frac{t-t^{*}}{10}\;\left(when\;0<=t<t^{*}+10\right)$$

*t*: time step of the detection point.

*t**: time step of the anomaly point.

As described above, in contrast to the change finder that detects changes using the TOPIX index itself, our method is considered effective in detecting changes in the overall TOPIX from the connections among various financial indices, thus capturing changes in individual connections before they are reflected in the overall index. In addition, in contrast to the change point, which detects changes as a graph from the daily relevance graph of financial indices, our method is effective in detecting changes in graphs related to finance because it partitions domains based on changes in graphs caused by explainable market changes.

To further validate the effectiveness of this methodology, the same settings were used for other time periods. Figure 6 shows the TOPIX index from 2000 to 2020. The three periods were 2001–2008, when the TOPIX index fluctuated due to government monetary policy and the global financial crisis; 2009–2012, when the index stagnated due to the recession; and 2013–2020, when the TOPIX index fluctuated due to Abenomics and COVID-19. Figure 7a–c shows the results of the ROC curves for each period.

Between 2001 and 2008 (Figure 7a), our method was accurate in the lower left part of the graph, where the false-alarm rate is low. As can be seen in Figure 6, this period includes the financial crisis and is a time of ups and downs in the TOPIX index, and it is considered that our method captured the changes with high accuracy. Between 2009 and 2012 (Figure 7b), there is no clear difference in the accuracy of the indicators, but this is because this is a period when market values change little, as can be seen in Figure 6. Between 2013 and 2020 (Figure 7c), our method was accurate as well. This period also includes the timing of important changes, and our method captured them. Overall, our method outputted a lower false-alarm rate when we relaxed the responsiveness of the indicator, which indicates that our method is able to separate the change points from the rest of the graphs and clearly identify the changes. Our method and model, which are based on structural changes in the market, are effective for non-COVID-19 periods as well.

### 5.3. Change Explanation

When we asked the investors’ experts if the results of this study could explain changes that affect their investment decisions, they commented that the stability of the relationship between stocks and bonds is important for asset management and can be used to judge their relationship, and that the relationship between stock and bond interest rates could be used to early detect when the relationship between stock and bond prices differs from the norm. They also commented that PDC had explanatory power over a period of time, leading from a little risk-off to the initial phase of change, such as COVID-19.

As a specific example of a possible strategy, it has been suggested that when the PDC is going up, stocks are going down; thus, the ratio of stocks to assets should be lowered. In addition, when only the 5- and 7-year government bond rates leave the cluster, such as on 3 March 2020, the stock market continues to fall, and interest rates stop falling. This could be attributed to the investors’ anxiety and panic over COVID-19 and their expectation that the recession will continue; thus, their strategy is to run to cash because all assets are not profitable.

Looking at the price movements of TOPIX and JGB interest rates (Figure 8), both assets are down up until 9 March 2020, but JGB interest rates have risen significantly since then. Since bond interest rates and bond prices are usually inversely correlated, bond prices are falling. This situation suggests the effectiveness of the strategy of fleeing to cash, as both stocks and bonds are falling, and no assets are rising.

Investors can devise specific strategies and explain market movements, as described above, because this method is modeled on actual structural changes in financial markets, and its fitting to financial market movements provides explanatory power when detecting changes. The explanation for the change—that uncertainty about COVID-19 caused all assets to fall, which was manifested in the structure of the clusters in the graph—was linked to investment strategies of running to cash. This result is something that has not been explained in other papers, which are overall overviews with indicators such as [3]. This shows that sequential change detection during major events such as COVID-19 is beneficial to investors.

## 6. Conclusions

In this study, the financial market was represented graphically, and changes in its structure were analyzed using change-detection indicators that used GBE and inter-domain linkage, which were devised from a model of the flow of change in the structure. We were able to detect major changes in the market caused by the series of events in COVID-19. In addition to detecting changes, we were able to present investors with materials on which to explain the changes, and they could actually think about their strategies by visualizing the changes graphically and dividing the data into domains for detailed analysis. The model of the flow of changes in the market was quantitatively confirmed to be in line with reality, and the analysis was based on real movements that led to concrete action plans by investors. This study developed a model based on investors’ practical needs, demonstrated its effectiveness, and had it evaluated by real investors. We verified that change detection, including the explanation of the change, extracts useful information for investors, as described in Section 5.3.

For future prospects, we would like to consider how to divide and visualize the data so that we can present detailed causal relationships when explaining the changes and determine how to use alternative data for analysis.

## Figures and Tables

**Figure 1 entropy-24-01726-f001:**
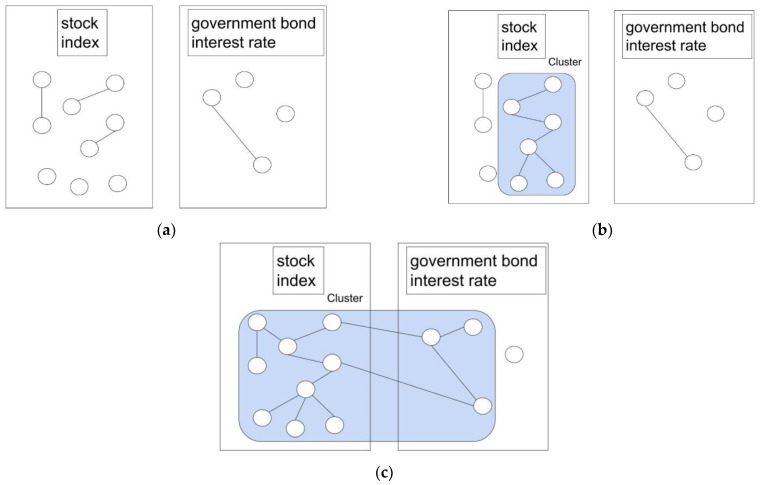
(**a**). Hypothetical graph of the normal state of the finance market made of two domains. (**b**). Hypothetical graph of the market beginning to be affected by a latent event. (**c**). Hypothetical graph of the market heavily affected.

**Figure 2 entropy-24-01726-f002:**
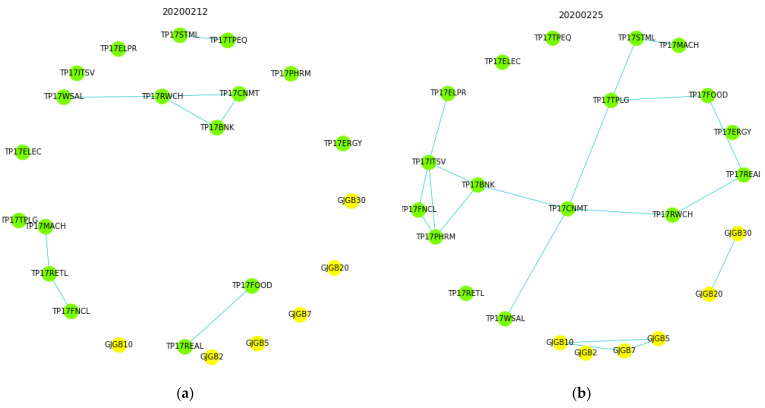
(**a**). Daily correlation graphs on 12 February 2020. (**b**). Daily correlation graphs on 25 February 2020. (**c**). Daily correlation graphs on 2 March 2020.

**Figure 3 entropy-24-01726-f003:**
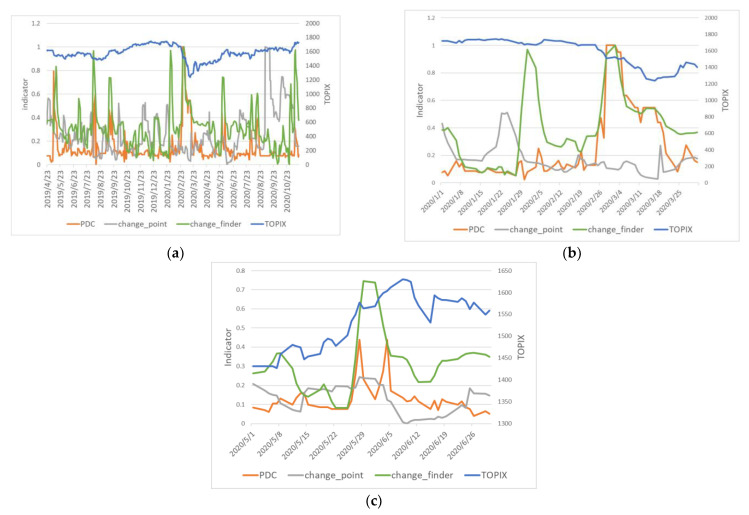
(**a**) Changes in change-detection indicators (threshold *θ* = 0.6). (**b**) Changes in change-detection indicators (threshold *θ* = 0.6, 2020/1~2). (**c**) Changes in change-detection indicators (threshold *θ* = 0.6, 2020/5~6).

**Figure 4 entropy-24-01726-f004:**
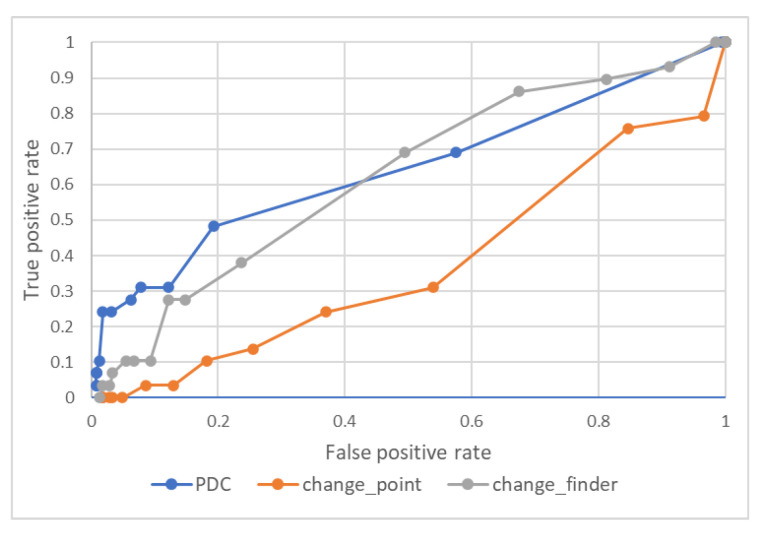
ROC curves for each change-detection indicator.

**Figure 5 entropy-24-01726-f005:**
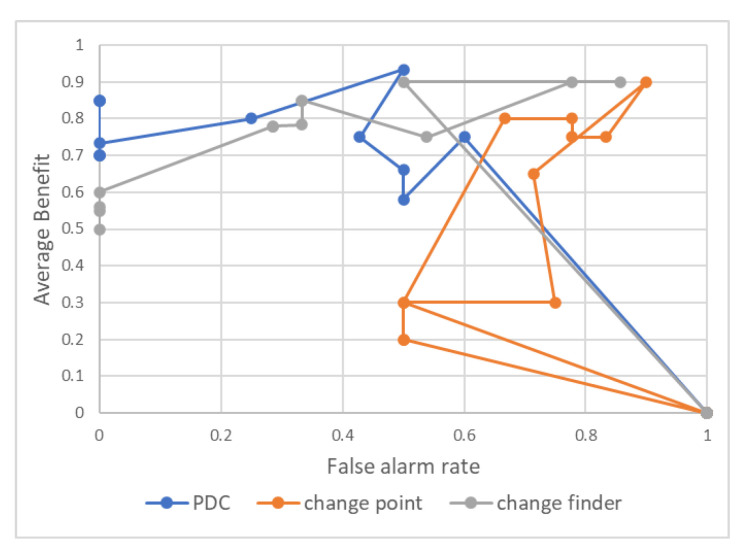
Average benefit–false-alarm rate for each change-detection indicator.

**Figure 6 entropy-24-01726-f006:**
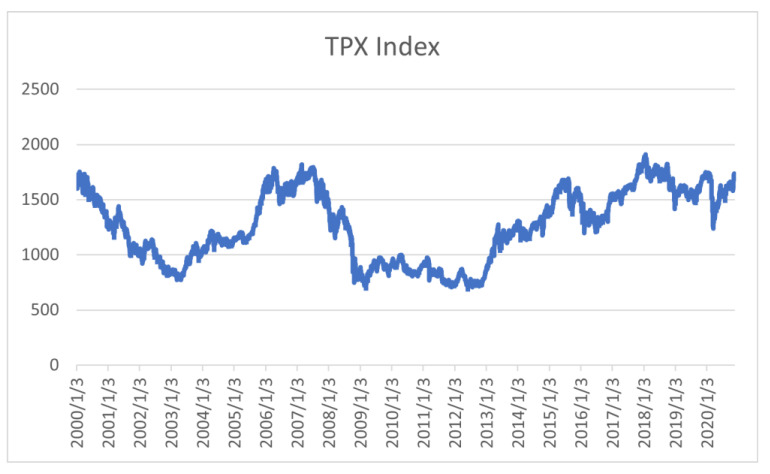
2000–2020 TOPIX index.

**Figure 7 entropy-24-01726-f007:**
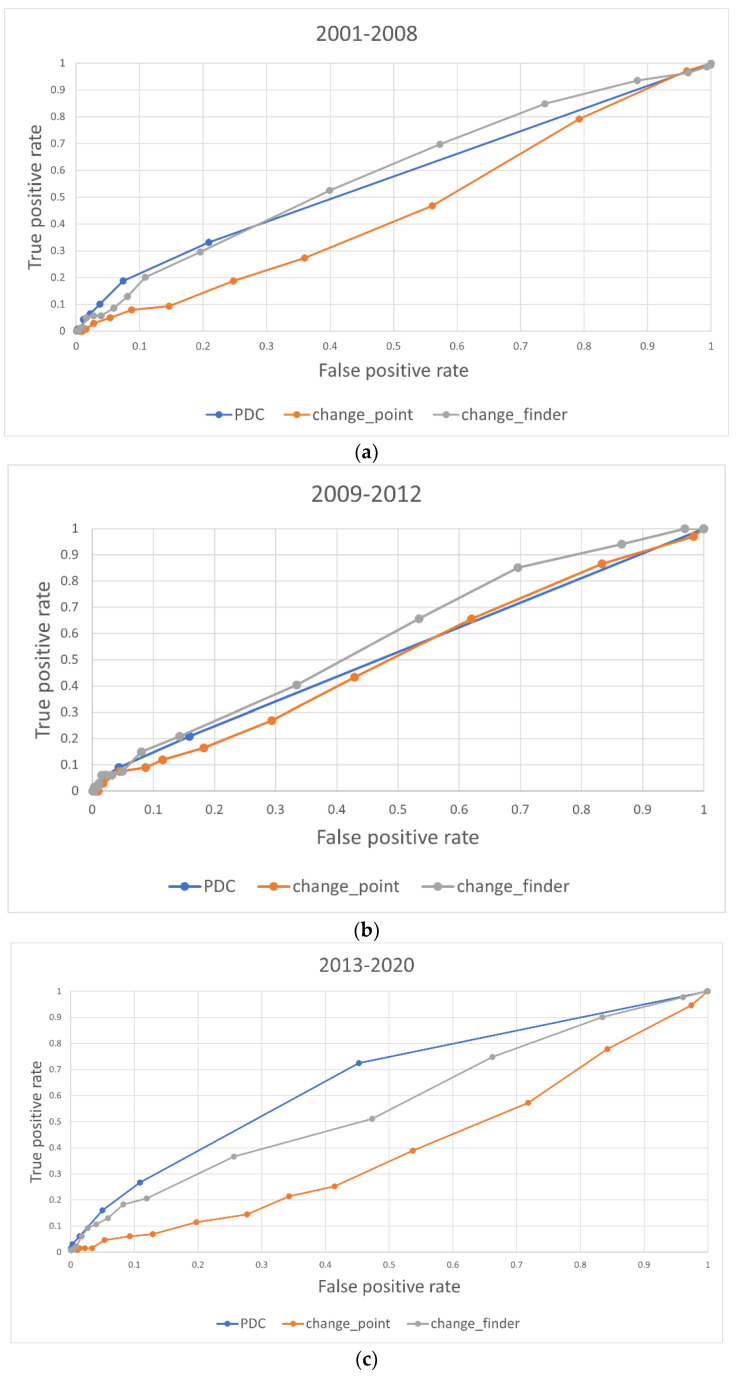
(**a**). 2001–2008 ROC curves for each change-detection indicator. (**b**). 2009–2012 ROC curves for each change-detection indicator. (**c**). 2013–2020 ROC curves for each change-detection indicator.

**Figure 8 entropy-24-01726-f008:**
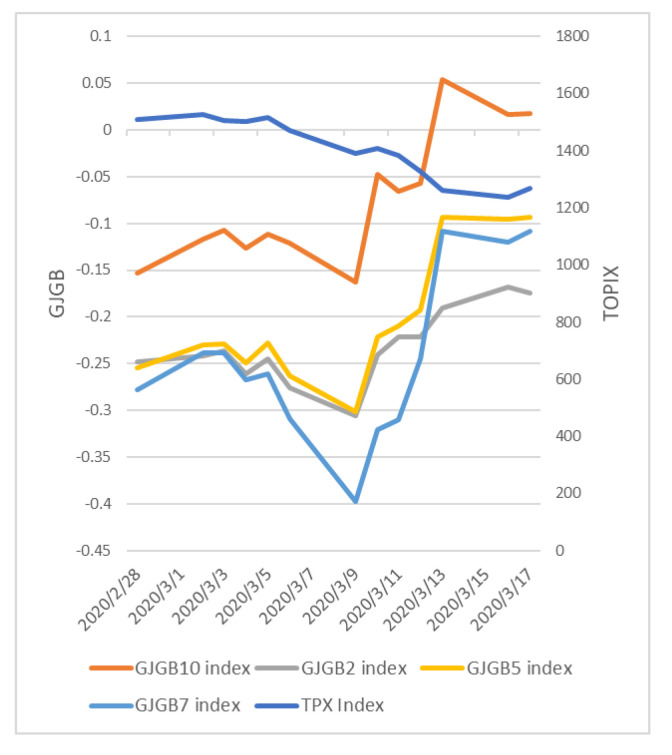
Price movements of TOPIX and JGB interest rates around March 2020.

## Data Availability

Not applicable.

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
