# Peer review of "Explanatory Change Detection in Financial Markets by Graph-Based Entropy and Inter-Domain Linkage"

_entropy, 2022, doi:10.3390/e24121726_

Round 1

Reviewer 1 Report

In this study, the aim is to design an original change detection indicator based on the structure of the graph. Results show that the original indicator was more effective than the comparison method in terms of both speed of response and accuracy. The paper has the potential to be accepted for publication. Before that, the authors are advised to consider the following comments and suggestions. Therefore, I recommend a minor revision for this submission.

1. The introduction part, I'm not know why it is important to explain that changes occur in financial markets. Maybe you can give us a more specific example to depict your ideas.

2. The introduction part, the literature review seems limited and requires more interpretation on why the Graph-Based Entropy and Inter-domain Linkage methods can be applied to detect changes in financial markets.

3. In the 3.3, why do you choose window width for the last 10 days? Have you do some sensitivity analysis? And, what is the sliding step of the window.

4. There is no more explanation for the theoretical basis of Eq. (3). Why can the indicator be characterized that changes in financial markets?

5. It is suggested that the same types of figures, combined into one figure, such as Fig.2, 3, 4, can be expressed as Fig.2(a), (b), (c), and Fig.5, 6, 7 similarly.

6. It is recommended that the figure explanation and conclusion be placed after the figure to enable the readers to clearly understand your intention.

7. The authors should lengthen the length of the data, e.g. by adding data before and after the financial crisis, to verify the accuracy of your proposed method.

8. The authors need to add practical and theoretical implications of their proposed work to this paper. Who are the main audiences of your research work?

Author Response

Thank you for your kind review. Please see the attachment.

Reviewer 2 Report

It is an interesting subject, but it will be necessary to make a definite statement to support the research theme, and it is regrettable that the contribution is not enough to make a paper based on this single content. Comments on the overall content are given below.

The manuscript lacks clarity in its presentation. This has hindered my understanding of the work. I will make observations and comments based on what I can decipher from the manuscript. Also, in order for this article to be published in a journal, you should have a clearer contribution. There is also a lack of literature on the latest research.

The author doesn't have enough explanation why he described the methodology to solve the problem. Before performing modeling, it is important to explain the rationale for this methodology based on existing literature. There are many things that are lacking in the methodological part. Authors should look into existing research in-depth on this and provide a methodology suitable for analysis.

Reviewer 3 Report

Authors of the manuscript “Explanatory change detection in financial markets by graph-based entropy and inter-domain linkage” analyze daily quotes of a set of the Japanese stock market indices and bond interest rates. They treat those assets as network nodes and apply several measures, including the graph-based entropy (proposed earlier by one of the Authors) and a so-called inter-domain linkage index (proposed in the current manuscript) in order to detect a potential structural change of the network related to the COVID-19 outburst in early 2020. The Authors claim that their approach enables for an early detection of changes which in their case consist of the emergence of coupling between the stock indices and the bonds.

The manuscript deals with an issue of the market structure change detection which is important from both the theoretical and (even more) practical points of view. Unfortunately, the manuscript lacks sufficient scientific soundness now. First, the Authors don’t provide  convincing arguments that the proposed methodology actually allows to extract more information from the data than other approaches based, for example, on the Pearson coefficient / correlation matrix calculated for the pairs of individual instruments or on the minimum spanning trees. The Authors are thus asked to provide more arguments in favour of their methodology.

Second, the present manuscript version gives an impression of being addressed to too narrow an audience, despite the fact that it has been submitted to a journal addressed to a fairly wide spectrum of readers. Among others, more emphasis needs to be put on defining the terms that may not be familiar to everyone (e.g., change finder and ROC).

Third, the graph-based entropy has to be fully defined with the appropriate formula and the advantage of using it has to be advocated. As the notion of entropy is the Journal highlight, it cannot be just mentioned without a broader presentation.

Fourth, the results have to be put in a context of the related research already published on the Covid-19 outburst and its influence on the financial markets of different types.

Of the finer points to mention:

  • Formula (2) lacks explanation of all the symbols it involves. The terms "L_n layers" and "lower domain" are also unclear there. The concept of layer does not appear anywhere in the work.

  • Variables should always be written in italics (e.g. page 2 below Eq. (1)).

  • The acronym DTW has to be introduced just after the first usage of the term it is related to.

  • The relations like that in line 148 have to be written as formulas not as a plain text.

  • The networks depicted in Figs. 2-4 are hardly readable on a hardcopy. Please use a larger font and mark heavier lines.

  • In lines 188-189 a phrase “begun to have an impact” is used twice and in each case it refers to a different date; please correct it.

The manuscript has to be improved along the above lines before it can be further processed.

Author Response

(The authors gave the same response as above.)

Round 2

Reviewer 2 Report

The contents were reflected as a whole and could not be edited well.

It seems that more improvement is needed to be published.

Reviewer 3 Report

Authors have more or less addressed all my concerns and now I find the manuscript more self-contained and its scientific context better highlighted.